# Computational Analysis of S1PR1 SNPs Reveals Drug Binding Modes Relevant to Multiple Sclerosis Treatment

**DOI:** 10.3390/pharmaceutics16111413

**Published:** 2024-11-03

**Authors:** Katarina Kores, Samo Lešnik, Urban Bren

**Affiliations:** 1Laboratory of Physical Chemistry and Chemical Thermodynamics, Faculty of Chemistry and Chemical Engineering, University of Maribor, Smetanova 17, SI-2000 Maribor, Slovenia; katarina.kores@um.si (K.K.); samo.lesnik@um.si (S.L.); 2Institute of Environmental Protection and Sensors, IOS, Beloruska 7, SI-2000 Maribor, Slovenia; 3Faculty of Mathematics, Natural Sciences and Information Technologies, University of Primorska, Glagoljaška 8, SI-6000 Koper, Slovenia

**Keywords:** molecular dynamics simulations, personalized therapy, single nucleotide polymorphisms, SNPs, linear interaction energy, LIE, multiple sclerosis, MS

## Abstract

**Background/Objectives:** Multiple sclerosis (MS) is an autoimmune disorder of the central nervous system (CNS) characterized by myelin and axonal damage with a globally rising incidence. While there is no known cure for MS, various disease-modifying treatments (DMTs) exist, including those targeting Sphingosine-1-Phosphate Receptors (S1PRs), which play important roles in immune response, CNS function, and cardiovascular regulation. This study focuses on understanding how nonsynonymous single nucleotide polymorphisms (rs1299231517, rs1323297044, rs1223284736, rs1202284551, rs1209378712, rs201200746, and rs1461490142) in the S1PR1’s active site affect the binding of endogenous ligands, as well as different drugs used in MS management. **Methods:** Extensive molecular dynamics simulations and linear interaction energy (LIE) calculations were employed to predict binding affinities and potentially guide future personalized medicinal therapies. The empirical parameters of the LIE method were optimized using the binding free energies calculated from experimentally determined *IC*_50_ values. These optimized parameters were then applied to calculate the binding free energies of S1P to mutated S1PR1, which correlated well with experimental values, confirming their validity for assessing the impact of SNPs on S1PR1 binding affinities. **Results:** The binding free energies varied from the least favorable −8.2 kcal/mol for the wild type with ozanimod to the most favorable −16.7 kcal/mol for the combination of siponimod with the receptor carrying the F205^5.42^L mutation. **Conclusions:** We successfully demonstrated the differences in the binding modes, interactions, and affinities of investigated MS drugs in connection with SNPs in the S1PR1 binding site, resulting in several viable options for personalized therapies depending on the present mutations.

## 1. Introduction

Multiple sclerosis (MS) is an autoimmune neurological inflammatory disorder of the central nervous system (CNS). It leads to myelin and axonal damage in the CNS [1,2]. The occurrence of MS in the world is rising, the highest being in the developed world [3]. The onset of MS is usually in adulthood, typically without apparent cause [2,3,4]. Currently, no known cure exists, but multiple disease-modifying treatments (DMTs) are available for different MS variants. DMTs exhibit different mechanisms of action. Among the most common is the modulation of Sphingosine-1-Phosphate Receptors (S1PRs) [3,4]. S1PRs belong to the GPCR superfamily of G protein-coupled receptors, and participate in a wide range of signaling pathways, most notably in immune responses, where they regulate the egress of lymphocytes from lymph nodes, thereby facilitating their circulation between the blood and lymphatic systems, which is essential for immune function; CNS, where they are involved in physiological functions, including the migration of neuronal progenitor cells, astrocyte communication, oligodendrocyte survival and myelination, microglial regulation, and maintaining the blood–brain barrier’s integrity; as well as in cardiovascular systems, where S1PRs regulate heart rate, vascular tone, blood pressure, vascular maturation, and play a significant role in endothelial cell function or the development of the vascular system [5,6]. The activation of S1PRs in immune cells plays an essential role in the regulation of cytokine release and (auto)-antibody production, which makes them primary drug targets in MS [5,7].

The S1PR1, a transmembrane protein, has seven transmembrane (TM) domains. A ligand-binding pocket is located at the extracellular portion of the receptor. Multiple amino acids are crucial in S1P binding [5], which raises the question what would happen if a mutation occurred at these positions. If this mutation is nonsynonymous, the binding pattern of an agonist/antagonist is likely to change, and the drug’s effectiveness may increase/drop [8,9]. Single nucleotide polymorphisms (SNPs) in the S1PR1 gene are common in the human population, so determining which ones are relevant in differentiating drug binding mechanisms may be crucial for understanding differential drug response. With the rapid development of pharmacogenomics, individuals’ biomarkers and mutations will be accessed more effectively and quickly, which may lead to personalized medicinal therapies based on individual’s SNPs [10,11].

Multiple FDA-approved drugs effectively manage different types of MS. They exhibit different mechanisms of action and effectiveness. Our research focused on four moderately effective drugs with S1PR1 affinity (Figure 1) that are used to treat relapsing forms of MS (RMS) [3,12,13,14]. The natural endogenic modulator of S1P_1–5_ receptors, sphingosine-1-phosphate (Figure 1a), represents a zwitterionic lysophospholipid that consists of a hydrophobic alkyl tail and a negatively charged polar head [15]. The fingolimod (S)phosphate (FTY720-P) (Figure 1b) forms the first generation of S1PR modulators that was approved for MS treatment. It acts as an agonist by inducing the G_i_ activation. However, because it is nonselective and binds to all S1PR subtypes, except for S1PR2, it exhibits many adverse side effects, most notably pulmonary epithelial leakage and brain edema [4,15]. Consequently, the second generation of RMS treatment drugs was introduced: siponimod (BAF312), ozanimod, and ponesimod (Figure 1c–e, respectively). Their selectivity is limited to S1PR1 and S1PR5, limiting unwanted side effects and reducing undesirable side effects [4,13,14,15].

In this work, we aim to determine the impact of different nonsynonymous mutations in the S1PR1 binding site on the binding of S1P and related drugs (Figure 1). Xu et al. [15], Parril et al. [16] and Fujiwara et al. [17] proposed several amino acids within the binding pocket of S1PR1 that can participate in the binding of S1P and related drugs. S1P has experimentally shown different binding affinities based on the mutations in the binding site [16,17], demonstrating that different mutations near the ligand can indeed cause binding alterations, making these studies well-suited for validating our LIE α and β parameters. To predict the affinities and binding modes of drugs based on the mutations in the S1PR1 binding site we implemented the linear interaction energy (LIE) method. The optimized empirical [18,19] *α* and *β* parameters were used on a set of known SNPs (Table 1) to determine the binding differences in the investigated drugs. We searched for SNPs that result in nonsynonymous mutations within the S1PR1 binding site and that could potentially change the binding pattern of S1P or the investigated drugs. We prepared a comprehensive list of SNPs in the S1PR1 (UniProt ID: P21453), regardless of their frequency in the population. From this list, we selected amino-acid residues located within 5 Å of S1P in the receptor binding site.

As seen in Table 1, the frequencies of the selected SNPs were very low in the general population. However, low-frequency and rare variants can still exert a major impact on disease [20,21], even when they are not prevalent in the general population. We did not possess data on the frequency of the selected SNPs, specifically in MS patients, and as such, we did not focus on population-level frequencies in our study. However, if these SNPs existed in a patient, they could significantly affect drug binding to the S1PR1, potentially altering treatment outcomes. Therefore, the utility of our research lies in its application for personalized treatment, providing insights relevant to individual patients rather than to broader populations. Although some of the investigated SNPs were predicted to have damaging effects on protein structure (categorized as “possibly damaging” or “probably damaging”), our research focused on their potential impact on drug binding rather than on inherent structural changes. This distinction was important because it aligned well with our aim to support personalized treatment strategies.

The M124^3.32^T mutation changed methionine’s longer hydrophobic side chain to a polar threonine, allowing potential hydrogen bonding. In V132^3.40^M, small valine became larger methionine. F205^5.42^L lost pi-stacking potential, possibly reducing binding affinity. T207^5.44^I swapped polar threonine for nonpolar isoleucine, eliminating hydrogen bonding. T211^5.48^P also disrupted hydrogen bonding, and A293^7.35^T replaced nonpolar alanine with polar threonine, capable of hydrogen bonding. A293^7.35^V showed a small increase in hydrophobicity. These mutations may directly affect binding or indirectly alter protein dynamics.

To determine which SNPs and corresponding protein mutations impacted the differentiation of drug-receptor binding, we combined extensive molecular dynamic (MD) simulations and LIE calculations to calculate the free energies of ligand–receptor binding. MD simulations proved valuable for researching binding mechanisms for drugs [22] and natural compounds [23]. We conducted extensive MD simulations of S1PR1 proteins with bound endogenic ligand or drugs (Figure 1) in combination with 15 mutations in the binding region of the receptor, resulting in ≥26 μs of simulation time. The LIE was used to compare how different mutations influenced the drugs’ binding. The method was previously successfully applied to predict the inhibitor affinity in Alzheimer’s disease target Aβ40 protofibril [24] and HIV-1 protease [25]; additionally, it was also used to provide mechanistic insights into the action of polyphenolic compounds [23].

The objective of this study was to examine the impact of specific S1PR1 SNPs on drug binding affinities, with the aim to support personalized treatment strategies for multiple sclerosis. This study’s workflow (Figure 2) involved initially selecting relevant SNPs based on predicted functional impacts, followed by computational modeling to construct both wild-type and SNP–mutant receptor structures. Molecular dynamics simulations were subsequently executed to observe the changes in receptor dynamics, and the binding affinities of MS drugs were calculated using the linear interaction energy methodology. This step-by-step approach offers insights into SNP-specific variations in drug interactions and provides a computational foundation for future experimental validation.

## 2. Materials and Methods

The structures of S1PR1 with co-crystalized S1P (PDB ID: 7VIE, chain D) and the investigated drugs, fingolimod (PDB ID: 7EO2, chain A), siponimod (PDB ID: 7TD4, chain D), and ozanimod (PDB ID: 7EW0, chain D) were downloaded from the RCSB Protein Data Bank (PDB). The system with ponesimod was prepared using a molecular docking procedure. We used the protein structure from the complex with co-crystalized S1P (PDB ID: 7VIE, chain D), where S1P served as a reference ligand for the molecular docking procedure performed with CmDock (https://gitlab.com/Jukic/cmdock (accessed on 2 February 2023)) [26]. The docking grid was applied as a sphere around the reference ligand with a 12 Å radius. During molecular docking, explicit waters were not considered. We executed 100 runs and subsequently obtained 100 potential docked poses of ponesimod. Based on the docking score, we selected the best docked pose to prepare the system with ponesimod for further simulations.

We applied CHARMM-GUI [27,28] for the preparation of simulation systems. S1PR1 with bound ligand was placed in a hydrated bilayer of POPC (1-palmytoyl-2-leoyl-*sn*-glycero-3-phosphatidylcholine lipids). Protein–ligand systems were prepared using the CHARMM36m force field [29] with WYF for an improved description of potential π-cation interactions and solvated within the cubic TIP3P water model box with the edge size of 125 Å and periodic boundary conditions. Subsequently, Na^+^ and Cl^−^ ions were added to achieve electroneutrality and the physiological 0.15 M neutralizing concentration. All the ligands were prepared to correspond to the physiological conditions at pH 7 (Figure 1). We also prepared systems with only ligands solvated by the TIP3P water and added physiological concentrations of ions for neutralization as needed by the LIE calculations, while running the MD simulations under identical conditions to the protein–ligand complexes. For each system, we combined the coordinate files of proteins and water molecules, then performed 50 steps each of both steepest descent and adopted basis Newton–Raphson energy minimizations to eliminate steric clashes and to optimize the atomic coordinates of the solute.

For MD simulations, the MD program NAMD [30,31] was used. The smooth Particle Mesh Ewald method [32] was applied to compute long-range electrostatic interactions. 225ps of heating and equilibration per system were performed with the HOOVER thermostat (constant particle number *N*, constant volume *V*, constant temperature *T*). The main production runs were performed for 100 ns per system with a 2fs time step starting after the final equilibration step. All the production MD simulations used the NPT ensemble (constant particle number N, constant pressure P = 1 bar, and constant T = 310.15 K). Five independent parallels of each system were produced, applying different random seed values, comprising ≥26 μs of MD simulations. All the simulated systems are collected in Table 2.

To apply the LIE method, we optimized its empirical *α* and *β* parameters for the simulation results of mutated systems to reproduce the experimentally determined *IC*_50_ values for S1P, fingolimod, and ponesimod. From measured *IC*_50_ values, the Δ*G*_exp_ values were calculated as
(1)$${∆G}_{exp}={RTlnIC}_{50},$$
where *R* is the universal gas constant, *T* is the temperature, and *IC*_50_ is the experimental inhibition constant of each drug. In the calculations of Δ*G*_exp_, we applied the temperature *T* = 310.15 K, and this was also the temperature used in our simulations. The values of Δ*G*_exp_ variated between −13 and −11 kcal/mol and were applied for determining the two empirical LIE parameters, *α* and *β*. We calculated the binding energies of all systems as
(2)$${∆G}_{bind}=\alpha \left({\langle E^{vdW}\rangle }_{bound}-{\langle E^{vdW}\rangle }_{free}\right)+\beta \left({\langle E^{elec}\rangle }_{bound}-{\langle E^{elec}\rangle }_{free}\right),$$
where *E^vdW^* and *E^elec^* represent van der Waals and electrostatic interaction energies between the ligand and surrounding atoms, which can be either solvent (free state) or solvated protein (bound state). 〈 〉 denotes the average of energies throughout the MD trajectory [18,33].

We conducted further in-depth analysis of the interactions between the ligands and the amino acids of the S1PR1 binding pocket using a Protein–Ligand Interaction Profiler (PLIP) [34]. We compiled all the interactions throughout all the five MD parallels for the wild-type and mutated systems. Then, we reviewed the occupancy of each interaction and prepared corresponding visual representations.

Finally, a redocking analysis was conducted to validate the docked pose of the ponesimod. We collected the protein structures used in the preparation of MD simulations and removed the co-crystallized S1P, fingolimod, siponimod, and ozanimod. We then redocked all the drugs back to their respective targets. The RMSD values between the co-crystallized and redocked poses of each drug were calculated to validate the method and the docked pose of the ponesimod.

## 3. Results and Discussion

The structures of the S1PR1 with different ligands remained stable during MD simulations, and no significant modifications to the protein structures or ligand positions were detected (Appendix A).

### 3.1. Empirical Parameter Optimization for LIE Calculations

To successfully use the LIE method to determine the influence of SNPs on drug binding, the empirical *α* and *β* parameters had to be optimized. First, we calculated the Δ*G*_exp_ values from the experimentally determined *IC*_50_ values (Table 3).

We collected van der Waals and electrostatic interaction energies from MD simulations for each parallel in the main simulations. The average interaction energies for each production run were calculated using Equation (2) (Appendix A), and the initial ΔG_bind_ was determined for each protein–ligand MD simulation using the average interaction energies of the non-bound ligand simulation. The empirical α and β parameters were initially set to 0.161 and 0.48, respectively [18,19]. We then performed parameter optimization, where we searched for the optimal α and β parameters that brought the computed data for wild-type S1PR1 with bound S1P, fingolimod, ozanimod closest to the experimentally determined values. The optimized empirical α and β parameters were 0.46 and 0.09, respectively. The low β value can be directly connected to the fact that we had bound zwitterions, charged molecules, and halogens. Almlöf et al. [19] described different models for determining β, where the authors showed that different hydrogen bond-donating groups lower the beta parameter. Similarly, Rifai et al. [44] and Ngo et al. [24] successfully applied the LIE method on SIRT1 and Aβ_40_ protofibril, respectively, with equally small or even negative β parameters. The average values of computed binding free energies for S1P, fingolimod phosphate, and ponesimod with wild-type S1PR1 were −13.76, −11.60, and −11.25 kcal/mol, respectively. Therefore, after minimizing the empirical parameters, the computed binding energies agreed well with the experimental values. To confirm our optimized empirical α and β parameters, we performed the validation with S1P bound to S1PR1, with different mutations within the binding site. Optimized α and β parameters were thus used for the ΔG_bind_ calculations of different S1PR1 mutants with S1P (Table 1) for method validation. Its results are presented in Table 4 with the average values of parallel MD simulations and the corresponding standard deviations.

Comparing our results with the experimental values from Parril et al. [16] and Fujiwara et al. [17], we can see that the N101I mutation was less favorable than the N101K mutation, as well as that the E121A mutation was less favorable than the E121Q mutation. The binding free energies of both the E121 mutants and R292 mutants were higher than of the WT; also, the E121 mutations were less favorable than the R292 mutations. This is all in agreement with the radioligand binding assays performed by Parril et al. [16]. The mutation W269E was more favorable than W269A with a large difference in the binding free energies, which again coincides with findings by Fujiwara et al. [17].

### 3.2. SNP-Based Binding Modes

Using the optimized parameters for LIE calculations, we compared the impact of the known SNPs in the binding site of S1PR1 (Table 5) on drug binding. In what follows, we focus on each drug individually and compare the impact of mutations on their binding to S1PR1.

The S1PR1 modulation regulated the recirculation of lymphocytes between the blood and lymphoid tissues. The binding of the modulator downmodulated the egress of the T and B cells from lymph nodes. Therefore, the stronger the binding of the modulator, the lower the inflammatory response, which formed a key mechanism in MS regulation [6,15].

As can be observed in Table 5, all the investigated mutations lowered the binding affinity of S1P, likely resulting in the impaired endogenic modulation of S1PR1. If we compare the binding affinities of drugs and the endogenic modulator S1P, we notice that while S1P is the most effective in combination with the wild-type form, the investigated drugs presented better calculated binding affinity when at least one SNP was present. This is an important finding, as we do not want S1P to severely interfere with the drug binding, since this could result in ineffective therapy. The siponimod exhibited the most favorable binding in all the simulated systems. However, since the success of a drug therapy in an individual depends not only on the binding free energy, but also on other factors, including pharmacokinetics and how the patient accepts the drug and its potential side effects, we, therefore, focused on comparing the impact of the mutations on each drug individually.

Analyzing the binding affinities of drugs according to different SNPs revealed that some mutation–drug combinations are more favorable than others, as one can observe from the plotted heatmap (Figure 3). The binding affinity of fingolimod phosphate was influenced by specific mutations, with minor variations observed compared to the wild type. Notably, the mutations T211^5.48^P, A293^7.35^T, and A293^7.35^V enhanced the binding of the drug relative to the wild type (Table 5). In contrast, siponimod, ozanimod, and ponesimod exhibited consistently improved binding across all the examined mutations, characterized by significantly lower binding energies. Among these, siponimod demonstrated the highest binding affinity in the presence of the F205^5.42^L, A293^7.35^T, T207^5.44^I, and A293^7.35^V mutations. Similarly, the ponesimod shared a binding mode analogous to siponimod, although with a slightly different preference for mutations. Both the siponimod and the ponesimod showed a reduced binding affinity to the wild type and the T211^5.48^P mutant. Ozanimod exhibited a different binding mode than the other modulators. The mutations V132^3.40^M, M124^3.32^T, and F205^5.42^L increased the binding affinity of ozanimod the most while the T207^5.44^I increased the binding affinity the least.

To sum up the overall results, if a patient would exhibit the M124^3.32^T or V132^3.40^M mutation, a reasonable choice would be ozanimod, because its binding free energy with these mutants was one of the most favorable. Moreover, if the F205^5.42^L mutation was present in the S1PR1 binding site, the siponimod would be a sensible choice, with ozanimod and ponesimod as potential substitutions. When the T207^5.44^I mutation occurs, preferred therapy drugs would be siponimod and ponesimod. In the case of the T211^5.48^P mutant, a favorable drug based on our calculations would be fingolimod phosphate. Moreover, when the mutants A293^7.35^T or A293^7.35^V occur, there are three options that could work well: siponimod, ponesimod, and fingolimod phosphate, with ozanimod also being a potential choice.

To further investigate the differences between binding modes, we prepared a detailed interaction analysis for all the systems reported in Table 5 using PLIP. For better clarity of the obtained results, we focused on the interactions with ≥25.0% occupancy throughout each MD simulation. The interaction analysis of all the systems is presented in Appendix A. However, to better explain the main findings emphasizing the critical impacts of SNPs on binding affinities and interaction types we here highlight two prototypical examples: first, the difference in S1P binding to wild types (the strongest binding) and to proteins with the A293^7.35^T mutation (the weakest binding); and the second, the difference in siponimod binding to wild types (the weakest binding) and to proteins with the A293^7.35^T mutation (one of the strongest bindings).The interaction analysis of S1P bound to the wild type (Figure 4a) and to the A293^7.35^T mutant (Figure 4b) revealed that S1P in the wild-type forms more hydrophobic interactions with multiple amino acids, including A293^7.35^. In the A293^7.35^T mutant protein, S1P did not interact with threonine293, likely due to steric hindrance from threonine’s longer side chain, which also obstructed the interaction with the polar head of the S1P. Additionally, the hydrogen bonding of S1P was stronger in the wild type compared to the A293^7.35^T mutant. These findings are crucial because they highlight how specific mutations can drastically alter the binding affinity of a natural ligand, potentially impacting a receptor’s normal function and the effectiveness of endogenous modulation. Figure 5 highlights the interaction analysis of siponimod binding to the wild type (a) and to the A293^7.35^T mutant (b). Siponimod forms more hydrogen bonds with the A293^7.35^T mutant protein. Moreover, the loss of hydrophobic interactions between siponimod and threonine, present in the mutant protein, was also evident. This is important, as it demonstrates that certain mutations can enhance drug binding through increased hydrogen bonding, while simultaneously losing hydrophobic interactions, which could inform the design of more effective therapeutic agents tailored to specific genetic profiles.

### 3.3. Ponesimod Pose Validation

The docked ponesimod pose was validated through a redock procedure, where we applied CmDock (https://gitlab.com/Jukic/cmdock (accessed on 2 February 2023)) [26] to redock S1P, and fingolimod, siponimod, and ozanimod to their S1PR1 crystal structures. The binding grid was determined with a 12 Å radius. Due to their flexibility, we considered the ten highest-scoring poses for each ligand and selected three poses per ligand with the lowest RMSD (Table 6). The RMSD values for S1P and fingolimod phosphate were higher due to their flexible alkyl tail, but the poses were still very similar (Figure 6). In contrast, the RMSD values for ponesimod and ozanimod were lower due to their more rigid structures (Figure 7).

## 4. Conclusions

In this study, we examined how nonsynonymous mutations in the S1PR1’s binding site affect ligand and drug binding in MS treatment by employing MD simulations and LIE calculations to predict binding affinities and guide personalized therapies. Structures of S1PR1 with co-crystallized S1P, fingolimod, siponimod, and ozanimod were obtained from the RCSB Protein Data Bank (PDB). The system with ponesimod was prepared via molecular docking applying the S1P structure as a reference ligand. Extensive MD simulations were performed using NAMD, encompassing over 26 μs of trajectories across various systems. LIE calculations were utilized to determine the binding free energies of the investigated compounds, with the parameter optimization based on experimentally determined *IC*_50_ values. The optimized α and β parameters were 0.46 and 0.09, respectively, and were subsequently applied to calculate the binding free energies of the investigated compounds in conjunction with different known mutations.

Our findings indicate that S1P binds most favorably to the wild-type S1PR1, while mutations reduce the affinity of this natural modulator. Conversely, these mutations positively affected drug binding. We demonstrated the potential for personalized therapy, as different drugs exhibited varying binding affinities depending on the mutations present. We proposed distinct personalized treatment options based on seven known SNPs in the S1PR1 binding site. In this study, we focused on analyzing the effect of individual SNPs on drug binding to the S1PR1. However, as there exists the possibility of multiple nonsynonymous SNPs in the binding site, our developed methodology using molecular docking, MD simulations, and optimized LIE parameters, is well suited to be also applied in such cases. Last but not least, redocking analyses validated the docked pose of ponesimod by comparing the RMSD values between co-crystallized and redocked positions of S1P, fingolimod, siponimod, and ozanimod.

The primary limitation of this study is its reliance solely on computational results without direct experimental validation. This reliance may impact the accuracy of the predicted effects of SNPs on drug binding affinities and receptor dynamics. Additionally, the limited availability of population-specific data on SNP prevalence in multiple sclerosis patients restricts our ability to assess the generalization of our findings. These factors may affect the broader applicability of our results. Future experimental studies, including in vitro and in vivo validations, are crucial to confirm our computational predictions and to explore their clinical relevance in personalized treatment strategies.

This study establishes an important foundation for understanding how specific SNPs can alter drug binding affinities and interactions with the S1PR1, thereby paving the way for personalized treatment strategies in multiple sclerosis. Serving as a critical initial computational step, our findings can guide future experimental research and clinical studies, ultimately combining computational predictions with real-world applications in precision medicine.

## Figures and Tables

**Figure 1 pharmaceutics-16-01413-f001:**
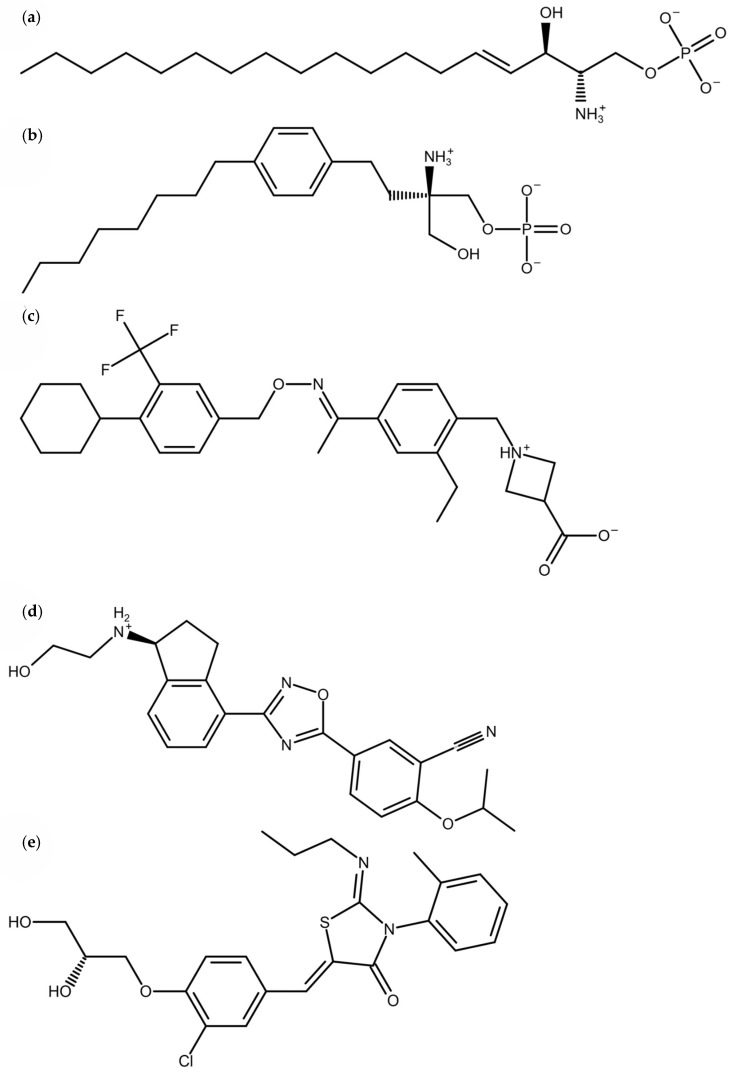
Chemical structures of the endogenic natural mediator and drugs used in MS treatment with a known S1PR1 action. (**a**) Sphingosine-1-phosphate (S1P); (**b**) fingolimod (S) phosphate (FTY720-P); (**c**) siponimod (BAF312); (**d**) ozanimod; and (**e**) ponesimod.

**Figure 2 pharmaceutics-16-01413-f002:**
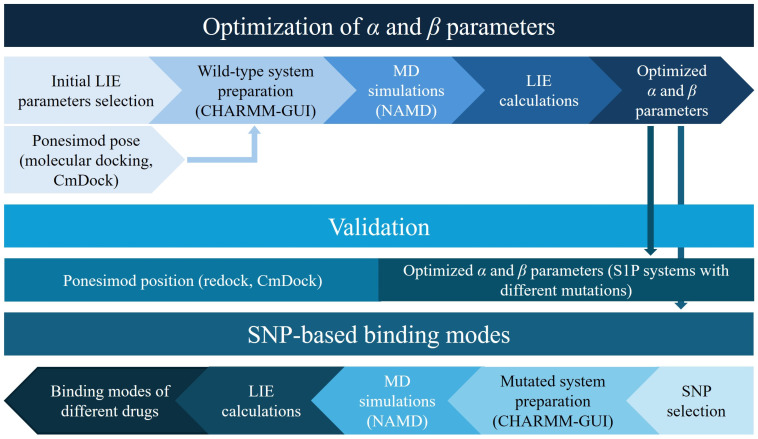
Workflow of this study. For the optimization of the empirical LIE parameters, wild-type systems with different drugs were prepared and subjected to MD simulations. The optimized α and β parameters were then used in subsequent steps: the validation of the parameters with MD simulations of the mutated systems bound with sphingosine-1-phosphate (S1P), the exploration of SNP-based binding modes where different SNPs were introduced, and finally, the performance of MD simulations for all the examined drugs.

**Figure 3 pharmaceutics-16-01413-f003:**
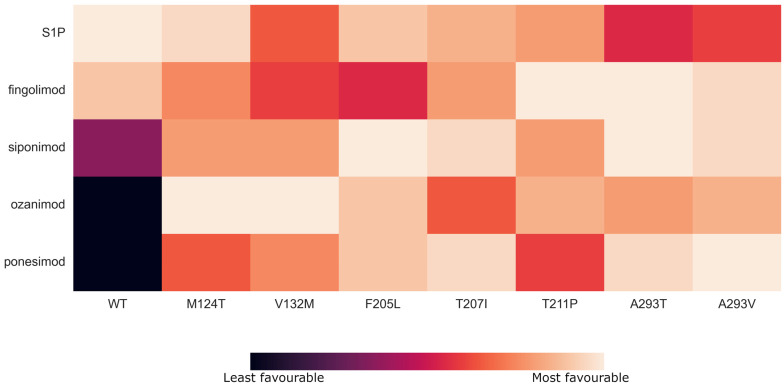
Heatmap representation of binding modes of investigated mutation–drug combinations. The values shown in the heatmap were calculated and colored according to the normalized binding free energy for each compound. Value 0 means that the binding free energy was the least favorable, while value 10 means that the compound had the most favorable binding free energy.

**Figure 4 pharmaceutics-16-01413-f004:**
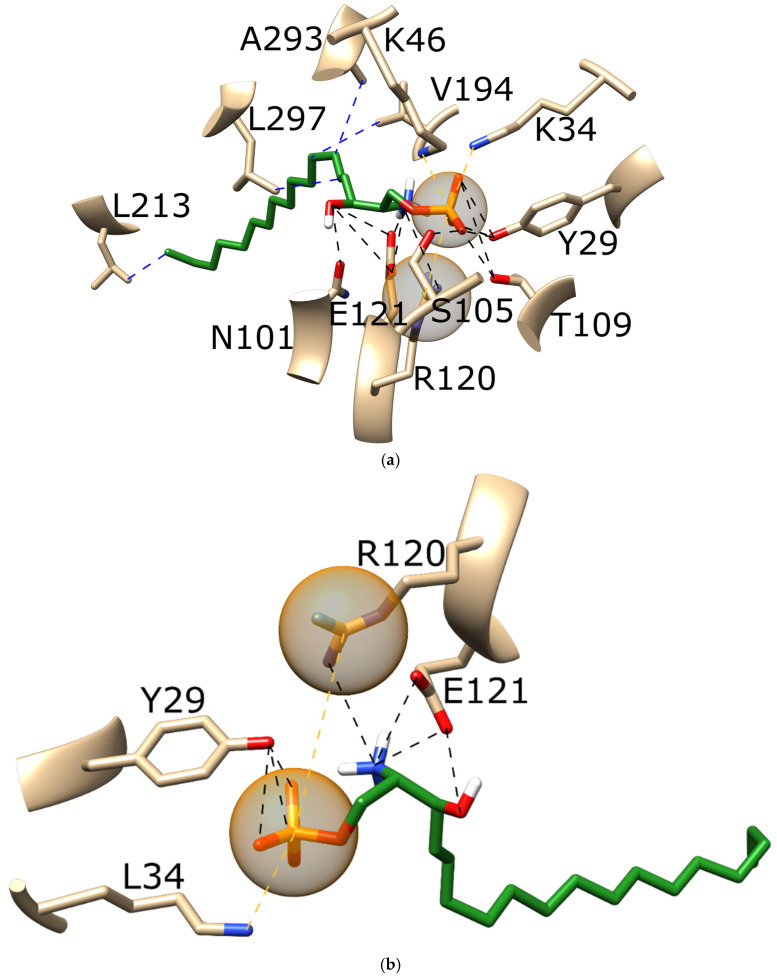
The interaction analysis of S1P (green) binding to (**a**) the wild type and (**b**) the A293^7.35^T mutant. Black dotted lines depict hydrogen bonds, blue dotted lines depict hydrophobic interactions, and orange dotted lines in combination with orange spheres depict salt bridges. For better clarity of the obtained results, we focused on the interactions with ≥25.0% occupancy throughout each MD simulation. A representative snapshot from the corresponding MD trajectory was applied. Implicit hydrogen atoms are not shown for better clarity.

**Figure 5 pharmaceutics-16-01413-f005:**
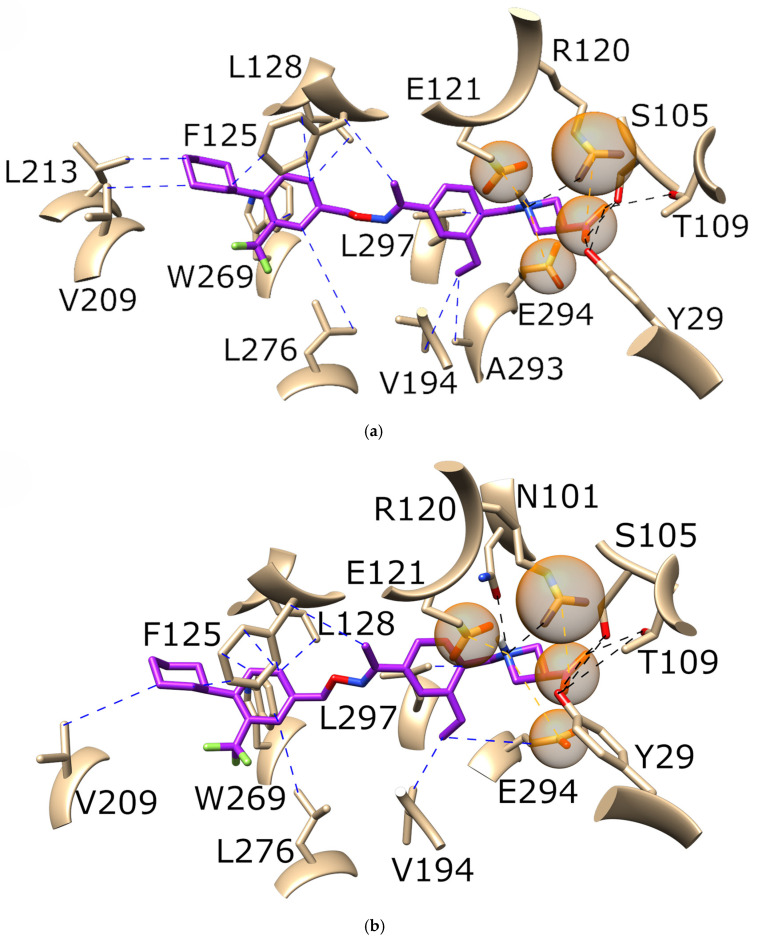
The interaction analysis of siponimod (purple) binding to (**a**) the wild type and (**b**) the A293^7.35^T mutant. Black dotted lines depict hydrogen bonds, blue dotted lines depict hydrophobic interactions, and orange dotted lines in combination with orange spheres depict salt bridges. For better clarity of the obtained results, we focused on the interactions with ≥25.0% occupancy throughout each MD simulation. A representative snapshot from the corresponding MD trajectory is applied. Implicit hydrogen atoms are not shown for better clarity.

**Figure 6 pharmaceutics-16-01413-f006:**
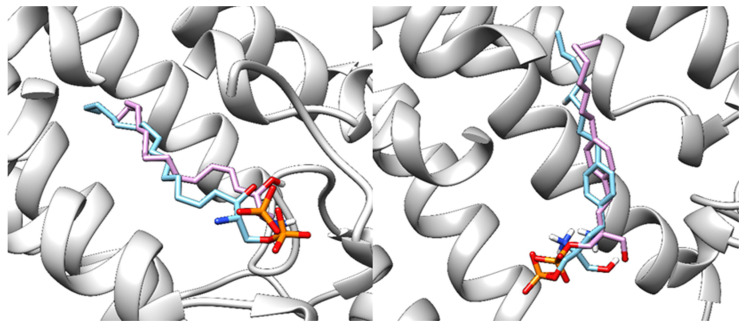
Native (blue) and redocked (light purple) poses of S1P (**left**) and of fingolimod (**right**) in S1PR1 protein (gray). RMSD: 2.41 and 2.46 Å, respectively.

**Figure 7 pharmaceutics-16-01413-f007:**
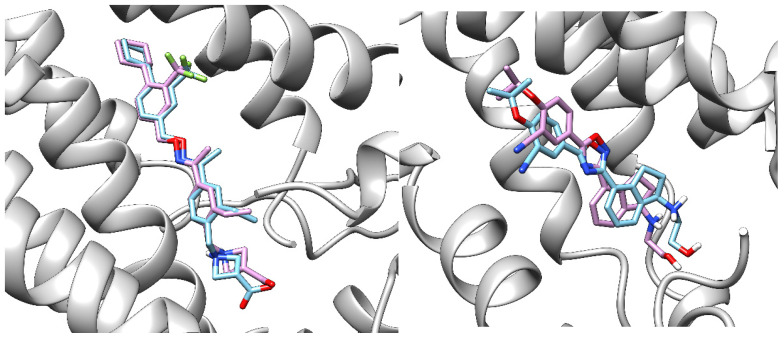
Native (blue) and redocked (light purple) poses of siponimod (**left**) and of ozanimod (**right**) in S1PR1 protein (gray). RMSD: 0.97 and 1.25 Å, respectively.

**Table 1 pharmaceutics-16-01413-t001:** Known reference SNPs with corresponding amino-acid variants; occurrence (frequency) in general population and potential impact.

SNPs			
Reference SNP	Variant	Frequency	Potential Impact
rs1299231517	M124^3.32^T	0.000004	possibly damaging
rs1323297044	V132^3.40^M	0.000004	probably damaging
rs1223284736	F205^5.42^L	0.000004	possibly damaging
rs1202284551	T207^5.44^I	0.00011	benign
rs1209378712	T211^5.48^P	0.000008	possibly damaging
rs201200746	A293^7.35^T	0.000004	benign
rs1461490142	A293^7.35^V	0.000007	benign

**Table 2 pharmaceutics-16-01413-t002:** Overview of prepared systems. For the LIE calculations the five systems with ligands in aqueous solution were prepared (non-bound ligand energies). For the optimization of the α and *β* parameters, the three systems with wild-type S1PR1 and S1P, fingolimod, and ponesimod ligand were prepared. An additional two with wild-type systems were prepared for the comparison of the mutation impact on the siponimod and ozanimod binding. The validation of the LIE method was performed with nine S1P systems. Thirty-five systems representing the most frequent SNPs were prepared in combination with S1P and all the investigated drugs.

Ligand	Variant	Purpose
S1P	water	To obtain non-bound ligand energies (LIE calculations)
fingolimod		
siponimod		
ozanimod		
ponesimod		
S1P	WT	Optimization of *α* and *β* parameters (S1P, fingolimod, ponesimod), comparison of the mutation impact (all)
fingolimod		
siponimod		
ozanimod		
ponesimod		
S1P	N101^2.60^I	Validation of optimized *α* and *β* parameters
	N101^2.60^K	
	E121^3.29^A	
	E121^3.29^Q	
	W269^6.48^A	
	W269^6.48^E	
	R292^7.34^A	
	R292^7.34^V	
	M124^3.32^T	Investigated SNPs
S1P	V132^3.40^M	
fingolimod	F205^5.42^L	
siponimod	T207^5.44^I	
ozanimod	T211^5.48^P	
ponesimod	A293^7.35^T	
	A293^7.35^V	

WT—wild type.

**Table 3 pharmaceutics-16-01413-t003:** Experimentally determined IC_50_ values for S1P, fingolimod phosphate, and ozanimod, and the corresponding calculated ΔG_exp_ values. We used average ΔG_exp_ values for S1P and fingolimod phosphate in the LIE optimization. A standard deviation (st. dev.) was also reported.

S1P		Fingolimod Phosphate		Ponesimod	
IC_50_ [M]	ΔG_exp_ [kcal/mol]	IC_50_ [M]	ΔG_exp_ [kcal/mol]	IC_50_ [M]	ΔG_exp_ [kcal/mol]
1.60 x 10^-10^ [35]	−13.90	2.80 x 10^-10^ [36]	−13.56	1.30 x 10^-8^ [37]	−11.19
4.70 x 10^-10^ [38]	−13.24	2.10 x 10^-9^ [39]	−12.31		
6.70 x 10^-10^ [40]	−13.02	2.20 x 10^-9^ [39]	−12.29		
1.40 x 10^-9^ [41]	−12.56				
1.40 x 10^-9^ [42]	−12.56				
1.40 x 10^-9^ [43]	−12.56				
average	−12.97	average	−12.72		
st. dev.	0.49	st. dev.	0.59		

**Table 4 pharmaceutics-16-01413-t004:** Computed results for method validation with S1P bound to mutated S1PR1. Mutations were selected based on the experimental findings by Parril et al. [16] and Fujiwara et al. [17]. For each mutant, the average binding free energy and its standard deviation were calculated. All the calculated free energies are collected in Appendix A.

Variant	Average Binding Free Energy [kcal/mol]	Standard Deviation
WT	−13.76	1.42
N101^2.60^I	−12.11	1.66
N101^2.60^K	−13.59	0.84
E121^3.29^A	−11.58	3.32
E121^3.29^Q	−12.24	3.11
W269^6.48^A	−11.18	0.94
W269^6.48^E	−14.15	2.14
R292^7.34^A	−13.48	1.72
R292^7.34^V	−13.23	1.77

WT—wild type.

**Table 5 pharmaceutics-16-01413-t005:** Calculated binding free energies of S1P and investigated drugs to the S1PR1 binding site in combination with different SNPs. The average free energy [kcal/mol] and the standard deviation were calculated for each variant. All the calculated interaction energies are collected in Appendix A.

Variant	S1P		Fingolimod		Siponimod		Ozanimod		Ponesimod	
	Average	St. dev.	Average	St. dev.	Average	St. dev.	Average	St. dev.	Average	St. dev.
WT	−13.76	1.42	−11.60	2.00	−14.62	0.37	−8.22	0.54	−11.25	0.40
M124^3.32^T	−13.56	1.03	−11.29	0.68	−15.99	0.62	−12.72	0.26	−13.18	0.50
V132^3.40^M	−12.82	0.85	−10.95	0.67	−16.12	1.01	−12.76	0.57	−13.47	0.46
F205^5.42^L	−13.51	1.54	−10.83	1.19	−16.72	0.49	−12.56	0.74	−13.90	0.95
T207^5.44^I	−13.29	0.98	−11.36	1.00	−16.55	0.55	−11.85	0.54	−14.06	0.32
T211^5.48^P	−13.24	0.43	−11.88	0.85	−15.98	0.79	−12.37	0.76	−13.12	0.46
A293^7.35^T	−12.50	1.40	−11.87	0.87	−16.71	0.92	−12.31	0.42	−14.09	0.56
A293^7.35^V	−12.67	0.78	−11.72	0.99	−16.52	0.44	−12.38	0.69	−14.21	0.52

WT—wild type.

**Table 6 pharmaceutics-16-01413-t006:** Energies and RMSDs of redocked S1P, fingolimod phosphate, siponimod, and ozanimod. Three poses with the lowest RMSD were selected among the ten highest-scoring poses.

S1P		Fingolimod		Siponimod		Ozanimod	
Energy [kJ/mol]	RMSD [Å]	Energy [kJ/mol]	RMSD [Å]	Energy [kJ/mol]	RMSD [Å]	Energy [kJ/mol]	RMSD [Å]
−12.52	3.43	−8.40	4.09	−24.00	1.09	−18.45	1.25
−11.98	3.32	−8.26	2.71	−22.95	0.97	−18.15	2.61
−11.54	2.41	−8.14	2.46	−21.07	1.32	−17.70	1.34

## Data Availability

All data generated or analyzed during this study are included in this published article.

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
