# Peer review of "Computational Analysis of S1PR1 SNPs Reveals Drug Binding Modes Relevant to Multiple Sclerosis Treatment"

_pharmaceutics, 2024, doi:10.3390/pharmaceutics16111413_

Round 1
Reviewer 1 Report
Comments and Suggestions for Authors
The title is too generic and should include the main findings of the study instead generally citing them.
The abstract is interesting, but lacks the proper identifications of tested polymorphisms studied. Additionally, numerical data should be included to provide a concrete demonstration of the findings.
Please, provide epidemiological data regarding the occurrence of SNP in the S1PR1 gene. The objective of the study should be cited in the last paragraph. I suggest that the authors reorganize the introduction section to improve comprehension regarding their goals. You should define SNP and the potential impacts of this occurrence. The text between lines 94 to 109 could be reduced.
The color of chemical groups should be standardized (Fig 1). Furthermore, authors are invited to include a figure containing the experimental flow of their study.
Please revise the misuse of abbreviations within the text.
I found the section on SPNs a little confusing. Since it's mentioned in the title, I expected it to be explained in more detail. Are the authors focusing solely on the mutations caused by the polymorphism? How were these mutations obtained and selected? Which sampled populations were used?
Some references are from the early 2000s. Are there any more recent publications available on the subject?
Authors are invited to include the main limitation of their study intending proper data interpretation.
How do you imagine that the data obtained in this study could be applied in a real world situation?
The data could be expressed differently to enhance comprehension. In the current form it is difficult to follow and understand.
Author Response
We thank the Reviewer for the careful reading of our manuscript as well as for his/her insightful suggestions on how to further improve its quality. Please see the attachment for our point-by-point responses and discussions of the highlighted questions.

Reviewer 2 Report
Comments and Suggestions for Authors
Review of the manuscript entitled
‘Single Nucleotide Polymorphisms in S1PR1 Receptor Alter Drug Binding in Multiple Sclerosis’ by Katarina Kores, Samo Lešnik, and Urban Bren
The authors in the manuscript entitled ‘Single Nucleotide Polymorphisms in S1PR1 Receptor Alter Drug Binding in Multiple Sclerosis’ provide an insight into the possible role of single nucleotide polymorphisms (SNP) in S1PR1 receptor which plays a role in the pathogenesis of multiple sclerosis (MS). The authors analysed all four drugs currently used against MS. This in silico study shows the potential of personalised therapy based on DNA sequence for MS patients. The literature review showed that the presented research has elements of novelty, however, some concerns need to be addressed before publishing.
1#
The authors include the list of studied SNPs and give the literature reference to it. However, I could not find the listed SNPs in the cited papers [15-16]. Please include the information on how the listed SNPs were chosen.
2#
The chosen SNPs belong to nonsynonymous group and thus could have an impact on the amino acid sequence in the protein. However, all analysed SNPs have highest minor allele frequency (MAF) below 0.01, according to Ensembl database. This means that the frequency of these SNPs is below 1% in the population and thus has a faint chance of entering clinical applications.
3#
The authors consider the occurrence of a combination of two SNPs in one patient in the Discussion section. I am wondering whether the molecular docking analyses allow for the calculation of the binding energy for a protein with a combination of two changes.
Author Response

(The authors gave the same response as above.)

Round 2
Reviewer 1 Report
Comments and Suggestions for Authors
The revision significantly improved the quality of the manuscript. My concerns were adequately addressed, and relevant modifications were performed. I thank you, the authors, for your consideration. I am now favorable to the acceptance of this manuscript as it is.
Reviewer 2 Report
Comments and Suggestions for Authors
I would like to thank the Author for the careful analysis of my suggestions on how to further improve its quality.